# Fermented Chinese Herbs Improve the Growth and Immunity of Growing Pigs through Regulating Colon Microbiota and Metabolites

**DOI:** 10.3390/ani13243867

**Published:** 2023-12-15

**Authors:** Junhao Zhang, Zhiheng Shu, Sixiao Lv, Qingwen Zhou, Yuanhao Huang, Yingjie Peng, Jun Zheng, Yi Zhou, Chao Hu, Shile Lan

**Affiliations:** 1College of Bioscience and Biotechnology, Hunan Agricultural University, Changsha 410128, China; 15197055901@stu.hunau.edu.cn (J.Z.); shuzhiheng@stu.hunau.edu.cn (Z.S.); lsx@stu.hunau.edu.cn (S.L.); shla4399@stu.hunau.edu.cn (Q.Z.); hyh0120@stu.hunau.edu.cn (Y.H.); zhengjun0120@stu.hunau.edu.cn (J.Z.); mystc@stu.hunau.edu.cn (Y.Z.); 2Guangdong Chuangzhan Bona Agricultural Technology Co., Ltd., Guangning 526339, China; hupengyingjie@163.com

**Keywords:** fermented digestion-promoting Chinese herbs, growing pig, growth properties, serum indicators, gut microbiota

## Abstract

**Simple Summary:**

Developing new antibiotic substitutes to promote pig growth and health, as well as new feed to improve feed utilization and reduce manure production during the breeding process, have become important ways to solve the current dilemma and promote the pig industry. In this study, we have documented that adding a 2% and 3% digestion-promoting fermented Chinese herbal formula to feed effectively promoted the growth of growing pigs, improved serum immune and antioxidant activities, ameliorated the structure of colonic microbiota, and altered the composition of metabolites in colon contents. In particular, adding a 2% digestion-promoting fermented Chinese herbal formula to feed effectively reduced opportunistic pathogens and increased potential probiotics in the colonic microbiota. This study provides experimental data for the application and feed development of digestion-promoting fermented Chinese herbs for pig production and expands the understanding of the impact of fermented Chinese herbs on the host’s gut microbiota and metabolism.

**Abstract:**

(1) Background: the development of new antibiotic substitutes to promote pig growth and health has become an important way to solve the current dilemma and promote the pig industry. (2) Methods: to assess the effects of a fermented Chinese herbal (FCH) formula on the growth and immunity of growing pigs, 100 Duroc × Landrace × Yorshire three-way crossed growing pigs were randomly divided into control and treatment groups that were fed a basal diet, and a basal diet with 1% (group A), 2% (group B), and 3% (group C) FCH formulas, respectively. A sixty-day formal experiment was conducted, and their growth and serum indices, colonic microbiota, and metabolites were analyzed. (3) Results: the daily gain of growing pigs in groups A, B, and C increased by 7.93%, 17.68%, and 19.61%, respectively, and the feed-to-gain ratios decreased by 8.33%, 15.00%, and 14.58%, respectively. Serum immunity and antioxidant activities were significantly increased in all treatment groups. Particularly, adding a 2% FCH formula significantly changed the colon’s microbial structure; the Proteobacteria significantly increased and Firmicutes significantly decreased, and the metabolite composition in the colon’s contents significantly changed. (4) Conclusions: these results indicate that the FCH formula is a good feed additive for growing pigs, and the recommended addition ratio was 3%.

## 1. Introduction

Pork is the second most preferred meat for consumption globally because of its low relative price [1,2,3]. In total, pork production is expected to grow by approximately 11 Mt by 2028 [3]. Large-scale farming is an important method for the sustainable development of the pig industry because it reduces the labor costs of raising pigs [4]. However, increased environmental and public concerns regarding manure management and welfare are expected to limit production growth [5,6]. Developing new antibiotic substitutes to promote pig growth and health, as well as new feed to improve feed utilization and reduce manure production during the breeding process, have become important ways to solve the current dilemma and promote the pig industry [7,8,9,10,11,12].

As an important component of traditional Chinese medicine, many Chinese herbal formulas with anti-inflammatory, antibacterial, immunomodulatory, and digestion-promoting functions have been developed [13,14,15,16,17,18]. For instance, traditional Chinese medicine theory suggests that *Radix Astragali* has anti-inflammatory effects [16], and four phenolic derivatives isolated from *Radix Astragali* exhibit potent inhibitory effects on tumor necrosis factor-α (TNF-α) production and TNF-α, cyclooxygenase-2 (COX-2), interleukin-1β (IL-1β), IL-6, and inducible nitric oxide synthase (iNOS) mRNA expression at 50 μM in lipopolysaccharide (LPS)-induced mouse peritoneal macrophages [19]. The essential oil from *Artemisia argyi* inhibits inflammation through the down-regulation of JAK/STAT signaling in LPS-induced RAW264.7 cells [20]. Hawthorn is rich in vitamins, quercetin, quercitin in flavonoids, maslinic acid, chlorogenic acid in organic acids, ursolic acid, and oleanolic acid in triterpenes [21], and has many bioactive functions, such as aiding digestion and antiatherosclerosis [22]. Dried tangerine peel is used to treat cough, indigestion, asthma, and other ailments. Many anti-inflammatory and anticancer flavonoids and essential oils have been isolated from dried tangerine peel [23].

Chinese herbs have been used to cure diseases for thousands of years. However, the bioactive ingredients in Chinese herbs are complex, and some natural Chinese herbal products cannot be directly absorbed by humans and animals. Moreover, the contents of most bioactive ingredients in Chinese herbs are low, and some natural products are toxic to humans and animals [24]. Fermentation is a Chinese herb processing technique based on the action of microorganisms under proper temperature, humidity, and moisture conditions that enhances their original properties and produces new effects, expanding their scope of application to meet clinical demands [25]. Furthermore, the fermentation of Chinese herbs can enhance Chinese herb bioactivities and decrease potential toxicities [24]. Recently, fermented Chinese herbs (FCHs) have been shown to exhibit growth-promoting, immune-enhancing, and disease-resistant properties [26,27,28].

The intestinal microbiota plays an important role in host growth [29], immunity [30,31], development [32,33], metabolism [34,35], and health [36,37]. Visconti et al. [38] estimated that the microbiome was involved in a dialogue between 71% of fecal and 15% of blood metabolites based on whole metagenome shotgun sequencing data in 1004 twins. Atarashi et al. [39] reported that the spore-forming component of indigenous intestinal microbiota, particularly clusters IV and XIVa of the genus *Clostridium*, promoted T_reg_ cell accumulation in mice colonic mucosa. They also found that the colonization of mice via a defined mix of *Clostridium* strains provided an environment rich in transforming growth factor-β, and affected Foxp3^+^ T_reg_ number and function in the mouse colon [39]. In promoting host growth, Qi et al. [40] reported that the body weight and lean mass of germ-free piglets were approximately 40% lower than those of normal piglets, and the deletion of the intestinal microbiota led to weakened muscle function and a reduction in myogenic regulatory proteins. Moreover, they also found that gut microbiota introduced into germ-free piglets via fecal microbiota transplantation not only colonized the gut but also partially restored muscle growth and development [40]. Furthermore, many probiotics that promote host growth or health have been isolated from intestinal microbiota, such as *Lactobacillus* strains [41,42] and *Bacillus* species [43,44]. Notably, the therapeutic effects of Chinese herbs on various diseases are related to the regulation of intestinal microbiota [45,46,47,48]. FCHs also promote growth and intestinal health and regulate the intestinal microbiota of weaned piglets [27,49] and chicks [50,51]. Therefore, we speculated that the addition of digestion-promoting FCHs to feed could improve the digestion, absorption rate, and growth of pigs, and this effect may be related to gut microbiota and metabolism. To test this hypothesis, in this study, we analyzed the effect of a digestion-promoting FCH formula on growth, serum biochemical indicators, colon microbiota, and the metabolome of the colon contents of growing pigs. This study provides experimental data for the application and feed development of digestion-promoting FCHs for pig production.

## 2. Materials and Methods

### 2.1. Preparation of FCH Feed

Dried hawthorn (*Crataegus pinnatifida* Bunge), dried tangerine peel, motherwort herb (*Leonurus japonicus* Houtt.), dried *Magnolia officinalis* bark, *Radix Astragali*, *Radix Cynanchum otophyllum* Schneid., folium artemisiae argyi, licorice (*Glycyrrhiza uralensis* Fisch.) root, and dandelion (*Taraxacum mongolicum* Hand.-Mazz.) was mixed in a ratio of 3:4:1:1:2:2:4:1:1. This was crushed and passed through a 60 mesh sieve. The water content was adjusted to 40%, inoculated with *Bacillus subtilis* (viable count ≥ 2.5 × 10^9^ CFU/g), placed in a fermentation bag, fermented for 5 days in an incubator at 37 °C, and finally dried for approximately 4 h at 60 °C to 10% moisture.

### 2.2. Animals and Experimental Design

This animal study was approved by the Biomedical Research Ethics Committee of Hunan Agricultural University (approval number: Lunshenke 2023 No. 105) and conducted in accordance with its guidelines.

A total of 100 Duroc × Landrace × Yorshire three-way crossed growing pigs (25.75 ± 0.14 kg) were divided into four groups with five replicates per group and five pigs per replicate according to the principle of consistent average body weight and sex. Pigs in the control group were fed a basal diet (Table 1), and the diets in groups A, B, and C were provided a basal diet with 1%, 2%, and 3% FCHs, respectively.

The study lasted for 65 days, including a pre-feeding period of 5 days, and was conducted at the experimental base of Chuangzhan Bona Agricultural Technology Co., Ltd. (Zhaoqing, Guangdong, China). During the experiment, the pigs were raised in the same ad libitum feeding environment. Immunization and disinfection were performed in strict accordance with pig farm management practices.

### 2.3. Determination of Growth Performance

The feed and remaining feed were weighed daily during the experiment to calculate the average daily feed intake (ADFI). The piglets with empty stomachs were weighed before feeding on the first and last mornings of the formal experiment. The average daily gain (ADG) and feed/gain ratio (F/G) were calculated based on the data collected in the experiment.

### 2.4. Determination of Serum Biochemical, Immune, and Antioxidant Indicators

At the end of the experiment, blood samples were collected from the ear vein of each pig using a vacuum tube. After standing at room temperature for 30 min, the sample was centrifuged at 3000 rpm at 4 °C for 15 min. The supernatant was then transferred to a polypropylene centrifuge tube and stored in liquid nitrogen. The serum total protein (TP), albumin (ALB), triglyceride (TG), glutamate oxaloacetate transaminase (GOT), alanine aminotransferase (ALT), total cholesterol (T-Chol), high-density lipoprotein cholesterol (HDL-Chol), low-density lipoprotein cholesterol (LDL-Chol), and blood urea nitrogen (BUN) were detected using corresponding kits (Nanjing Jiancheng Bioengineering Institute, Nanjing, Jiangsu, China). The serum immunoglobulin A (IgA), immunoglobulin M (IgM), immunoglobulin G (IgG), lysozyme (LZM), total antioxidant capacity (T-AOC), glutathione peroxidase (GSH-Px), total superoxide dismutase (T-SOD), and malondialdehyde (MDA) levels were measured using ELISA kits (Cusabio, Wuhan, Hubei, China).

### 2.5. Colon Microbiota Analysis

The colon contents of the growing pigs were collected for 16S rRNA sequencing. The microbial DNA was extracted using a fecal genome DNA extraction kit (Tiangen, Beijing, China), and its concentration and purity were determined using a NanoDrop2000 ultramicro spectrophotometer (Thermo Fisher Scientific, Waltham, MA, USA). The prokaryotic-specific primers of the 16S rRNA gene V3-V4 region were selected for PCR amplification of the colon contents [52]. Novaseq6000 (Illumina, Shanghai, China) was used for sequencing.

The raw reads were merged using FLASH 1.2.11. Quality control was conducted using Trimmomatic 0.3. UCHIME 8.1 was used to remove the chimera sequences to obtain high-quality sequences. QIIME2 2020.6 was used to evaluate alpha diversity, beta diversity, and the taxonomy of the colon microbiota.

### 2.6. Non-Target Metabolomics Analysis

The colon content metabolomes were determined using the UHPLC-QTOF-MS method on the LC-QTOF platform from Biomarker Technology Co., Ltd. (Beijing, China). Chromatography was performed using a Xevo G2-XS QT high-resolution mass spectrometer (Waters, Milford, CT, USA) with a 1.8 μm 2.1 × 100 mm Acquity UPLC HSS T3 column (Waters, Milford, CT, USA). The raw data were obtained using MassLynx V4.2 (Waters, Milford, CT, USA). Progenesis QI (Waters, Milford, CT, USA) was used for the peak extraction and peak alignment of the raw metabolomic data. The theoretical fragments were identified based on the online METLIN database of the Progenesis QI version 4.0 software and the self-built database of Biomarker Technology (Beijing, China). The metabolic pathway enrichment analysis and functional annotation were performed using the Kyoto Encyclopedia of Genes and Genomes database (KEGG) (https://www.genome.jp/kegg/ (accessed on 20 September 2023)).

### 2.7. Data Analysis

The data are presented as the mean ± standard error. Bartlett’s homogeneity test of variance and a one-way ANOVA were conducted using SPSS Statistics 26.0 (IBM Corporation, Armonk, NY, USA). Waller-Duncan, LSD, and Tookey s-b post-hoc tests were used for multiple comparisons analysis. The statistical significance was set at *p* < 0.05. Histograms and boxplots were obtained using R 4.2.3 with the ggpubr package. Spearman’s correlation coefficient was used to analyze the relationship between the colon microbiota and metabolites, and *p* < 0.05 and R > 0.60 was considered significant correlation.

## 3. Results

### 3.1. Effects of FCHs on the Growth Performance and Serum Physiological and Biochemical Indicators of Growing Pigs

There was no significant difference in the IBW of growing pigs among the groups (F = 1.036, *p* = 0.398; Figure 1A). Although there was no significant difference in the ADFI of growing pigs among the groups during the experiment (F = 0.531, *p* = 0.662; Figure 1D), the ADGs of growing pigs in groups A, B, and C significantly increased by 7.93%, 17.68%, and 19.61%, respectively, compared with the controls. The FBWs and ADGs of groups B and C were significantly higher than those of the control (*p* < 0.05; Figure 1B,C). Moreover, the F/G ratios in groups A, B, and C decreased by 8.33%, 15.00%, and 14.58%, respectively, compared to the controls, and the F/G ratios in groups B and C were significantly lower than those in the control (F = 5.847, *p* < 0.05; Figure 1E). However, the FBW, ADG, and F/G were not significantly different between groups B and C (*p* > 0.05; Figure 1).

The levels of serum TP, ALB, and HDL-Chol in all treatment groups were significantly increased compared to the controls (*p* < 0.01; Figure 2A,B,G), whereas the serum BUN, TG, LDL-Chol, GOT, and ALT levels were significantly decreased (*p* < 0.01; Figure 2C,D,F,H,I). However, the serum T-Chol levels were not significantly different (*p* > 0.05; Figure 2E). The levels of serum IgA, IgG, and IgM in all treatment groups were significantly higher than those in the control group (*p* < 0.05; Figure 2J–L). However, there was no significant difference in LZM activity (*p* = 0.278; Figure 2M). The activities of serum T-AOC, GSH-Px, and T-SOD in the treatment groups were significantly increased compared to those in the controls (*p* < 0.01; Figure 2N–P), whereas the MDA content was significantly decreased (*p* < 0.01; Figure 2Q). No significant difference was observed between groups B and C (*p* > 0.05; Figure 2N,P,Q), except for GSH-Px activity (*p* < 0.05; Figure 2O).

### 3.2. Effect of FCHs on the Colonic Microbiota of Growing Pigs

To assess the effect of FCHs on the colonic microbiota of growing pigs, the colonic microbiota in group B and the control group were analyzed. A total of 1,573,864 (157,386.4 ± 6517.70 per sample) effective merged tags were obtained from ten samples (five samples for each group). A total of 5453 amplicon sequence variants (ASVs) were detected. The rarefaction curve reached a plateau after sequencing more than 100,000 reads per sample, indicating sufficient sequencing data (Figure 3A). The abundance-based coverage estimators (ACE), Chao1, Shannon, and Simpson indices in group B were significantly elevated compared with those in the control group (*p* < 0.05: Figure 3B–E). The principal coordinates analysis indicated that the microbial structures in the colon were significantly different between the treatment and control groups (Figure 3E).

At the phylum level, Firmicutes, Bacteroidetes, Proteobacteria, Spirochaetota, Campylobacterota, Actinobacteriota, Desulfobacteriota, Verrucomicrobiota, Fibrobacterota, and Patescibacteria were the top 10 phyla in the relative abundance of the colon microbiota of growing pigs (Figure 3F). The relative abundances of Bacteroidetes, Campylobacterota, Desulfobacterota, Fibrobacterota, Patescibacteria, Proteobacteria, and Spirochaetota in group B were significantly higher than those in the control group (*p* < 0.001; Appendix A), whereas the relative abundances of Firmicutes, Actinobacteria, and Verrucomicrobia were significantly lower (*p* < 0.01; Appendix A).

At the genus level, 103 dominant genera were identified (Appendix A), and the LEfSe results indicated that the relative abundances of *Desulfovibrio*, *Anaerovibrio*, Prevotellaceae UCG 001, *Parabacteroides*, the Family XIII AD3011 group, the Eubacterium nodatum group, the Prevotellaceae NK3B31 group, *Treponema*, *Campylobacter*, and *Phascolarctobacteriumin* in group B significantly increased compared to the control group, whereas the relative abundances of *Streptococcus*, *Terrisporobacter*, UCG 005, *Ruminococcus*, *Escherichia_Shigella*, *Subdoligranulum*, *Roseburia*, *Clostridium sensu* stricto 1, the *Eubacterium ruminantium* group, and *Lactobacillus* were significantly decreased (Figure 3G,H).

### 3.3. Effect of FCHs on the Metabolomes of the Colonic Contents of Growing Pigs

The PCA results based on the metabolomes of the colonic contents indicated that the FCHs significantly changed the colon content metabolomes in growing pigs (Figure 4A). In the default mode, based on the fold change and *p* values, a total of 2594 metabolites with significant differences were identified in the colonic content samples (Figure 4B and Appendix A). Significantly different metabolites in lysine degradation, glyoxylate and dicarboxylate metabolism, D-amino acid metabolism, nicotinate and nicotinamide metabolism, carbon fixation pathways in prokaryotes, biosynthesis of plant secondary metabolites, protein digestion and absorption, and alpha-linolenic acid metabolism were significantly increased in the treatment group compared with the controls. However, the metabolites involved in glucosinolate biosynthesis, purine metabolism, the biosynthesis of various antibiotics and alkaloids derived from histidine and purine, histidine metabolism, toluene degradation, and biosynthesis of phenylpropanoids were significantly decreased in the treatment group (Figure 4C). In particular, the top 10 enriched significantly different metabolites were secologanin, 4-amino-4-deoxyarabinose, hexamethylpropylene amine, 5-methyl-5,6,7,8-tetrahydromethanopterin, PS(20:0/22:5(4Z,7Z,10Z,13Z,16Z)), methyl 2-(10-heptadecenyl)-6-hydroxybenzoate, N1-acetyl-tabtoxinine-beta-lactam, 10-propyl-5,9-tridecadien-1-ol, 5′’-phosphoribostamycin, and narbonolide, and the top 10 decreased significantly different metabolites were 4-(beta-acetylaminoethyl)imidazole, 3-methylbutyl 3-oxobutanoate, N-butyl-N-(4-hydroxybutyl)nitrosamine, 1-(3-aminopropyl)-pyrrolinium, CDP-DG(i-12:0/6 keto-PGF1alpha), PIP(22:4(7Z,10Z,13Z,16Z)/PGF1alpha), 1,6-dihydroxy-5-methylcyclohexa-2,4-dienecarboxylate, (1S,2R,3S,7R,10S,13S,14R)-1-methyl-14-propan-2-yl-2-propyl-12-azapentacyclo[8.6.0.02,13.03,7.07,12]hexadecane, UDP-2,3-diacetamido-2,3-dideoxy-alpha-D-glucuronate, and procaine (Figure 4D). It was also noteworthy that the propionic acid, heptanoic acid, caproic acid, stearidonic acid (SDA), and citric acid contents in group B were significantly higher than those in the controls (Appendix A).

According to the analysis of the metabolites closely related to major metabolic pathways, succinyl-CoA, L-malic acid, L-tyrosine, citric acid, L-tryptophan, L-serine, L-leucine, (−)-jasmonic acid, and alpha-linolenic acid were the metabolites linked to phenylalanine metabolism, the biosynthesis of phenylpropanoids and plant secondary metabolites, protein digestion and absorption, and alpha-linolenic acid metabolism (Figure 4E). Except for *Anaerosporobacter*, *Blautia*, *Fournierella*, *Firsingicoccus*, *Paludicola*, Prevotellaceae UCG 003, and UCG 009, the other dominant genera showed significant correlations with the differential metabolites (Spearman’s correlation coefficient > 0.6, *p* < 0.05; Figure 4F and Appendix A). It is noteworthy that the correlation pattern between those metabolites and the genera including *Acetitomaculum*, *Agathobacter*, *Akkermansia*, *Catenisphaera*, the Christensenellaceae R 7 group, *Clostridium sensu stricto* 1, *Coprococcus*, *Escherichia_Shigella*, *Faecalibacterium*, *Holdemanella*, the Lachnospiraceae AC2044 group, the Lachnospiraceae NK4A136 group, *Lactobacillus*, *Monoglobus*, *Peptococcus*, *Romboutsia*, *Roseburia*, *Ruminococcus*, *Solobacterium*, *Streptococcus*, *Subdoligranulum*, *Terrisporobacter*, *Turicibacter*, UCG_005, UCG_008, the *Eubacterium eligens* group, the *Eubacterium ruminantium* group, and the *Eubacterium xylanophilum* group was almost opposite to the correlation pattern between the metabolites and genera including *Anaerovibrio*, *Bacteroides*, *Campylobacter*, *Candidatus Saccharimonas*, *Candidatus Soleaferrea*, Clostridiales bacterium 42_27, *Colidextribacter*, *Desulfovibrio*, *Dorea*, the Family XIII AD3011 group, Family XIII UCG 001, *Fibrobacter*, *Incertae Sedis*, *Lachnospira*, Lachnospiraceae UCG_010, the Lachnospiraceae XPB1014 group, the *Megasphaera*, NK4A214 group, *Negativibacillus*, *Oscillibacter*, *Oscillospira*, *Parabacteroides*, *Phascolarctobacterium*, *Prevotella*, the Prevotellaceae NK3B31 group, Prevotellaceae UCG_001, Prevotellaceae UCG_003, *Quinella*, the Rikenellaceae RC9 gut group, *Sphaerochaeta*, *Treponema*, UCG_001, UCG_002, the *Eubacterium fissicatena* group, the *Eubacterium nodatum* group, the *Eubacterium siraeum* group, and the dgA_11 gut group (Figure 4F and Appendix A).

## 4. Discussion

Fermented feed is beneficial to the growth and production of pigs [53], and fermented feed additives have a positive effect on the intestinal microbiota of pigs by promoting the proliferation of beneficial bacteria and reducing the number of potentially harmful bacteria [54]. The reduction in pH after the fermentation of compound Chinese herbs may be beneficial for inhibiting the growth and colonization of pathogens [55]. Our results indicated that adding different doses of FCHs to the basal diet improved the growth performance and feed efficiency of growing pigs. The ADG of growing pigs increased with increasing FCH content, and the ADFI also showed an increasing trend, consistent with the results of Shi et al. [56]. The improvement in growth performance may be due to the optimization of feed palatability and the improvement in intestinal enzyme activity and feed efficiency after the fermentation of Chinese herbs [57].

The serum TP, ALB, and BUN levels are used to indicate the body’s protein metabolism and synthesis ability, which are closely related to the body’s growth performance. Our results showed that adding 2% FCHs to the basal diet significantly increased serum TP and ALB and decreased the serum BUN concentrations, consistent with the results of Sun et al. [58]. The main function of serum LDL-Chol is to transfer endogenous cholesterol to the extrahepatic tissues. The main function of serum HDL-Chol is cholesterol reverse transport, which can transport cholesterol from extrahepatic tissues to the liver for metabolism and is then excreted through the bile. The increase in LDL-Chol concentration increases the risk of coronary heart disease and directly promotes atherosclerosis formation [59]. Our results showed that the levels of serum LDL-Chol and HDL-Chol in growing pigs in each treatment group were significantly improved compared to those in the control group. The GOT and ALT levels were negatively correlated with liver and cardiac functions. Elevated levels of these two enzymes reflect organ injury or disease, and a large number of intracellular enzymes enter the blood, resulting in elevated serum enzyme activities [60]. Our results indicated that the serum GOT and ALT levels of growing pigs in each treatment group were significantly reduced, indicating that consumption of FCHs had a protective effect on the liver and heart. These results are consistent with the report by Parimoo et al. [61].

The serum antioxidant index is an important indicator of animal health [62]. Dowarah et al. [63] proved that the improvement in the nutrient digestibility and growth performance of pigs was closely related to the improvement in antioxidant status. The use of *Bacillus licheniformis* to process *Carthamus tinctorius* L. can hydrolyze the glycosides to aglycogens, which significantly improves the body’s ability to eliminate hydroxyl radicals, and inhibits the production of liver oxides, and the symptoms of erythrocyte hemolysis [64]. T-SOD and GSH-Px are important antioxidant enzymes that help cells resist free radical damage [65]. T-AOC represents the total antioxidant level composed of various antioxidant substances and enzymes, indicating its ability to resist oxidative damage caused by reactive oxygen species [66]. MDA is naturally produced in vivo as a final product of membrane lipid peroxidation, and its elevated concentration indicates cell membrane damage [67]. Moreover, the medicinal active substances of Chinese herbs can promote lymphocyte synthesis and improve immunoglobulin levels and phagocytic ability [68]. Serum IgA, IgG, and IgM are important components of mammalian humoral immunity, which are the main immunoglobulins that protect the body against pathogen and virus invasion [69]. Serum LZM is a natural “bacteriostatic agent” of the body. As a part of the immune defense mechanism in the body, it participates in a variety of nonspecific immune responses, cracking the peptidoglycan chemical bonds of bacterial cell walls and causing them to break and dissolve to achieve bactericidal and anti-inflammatory effects [70]. Additionally, LZM exhibits immunomodulatory properties by regulating IgA production in the intestinal mucosa and serum of pigs [71]. Our results showed that dietary supplementation with 2% or 3% FCHs significantly increased serum T-SOD, GSH-Px, T-AOC, IgA, IgG, and IgM levels, and significantly decreased the serum MDA concentration in growing pigs. These results indicate that adding FCHs to the basal diet of growing pigs improves serum antioxidant, biochemical, and immune indices, which is beneficial to the health of growing pigs.

A healthy intestinal microbiota is important for the intestinal absorption and development of the immune system of growing pigs [72]. Bunte et al. [73] reported that feeding pigs with fermented feed improved the α-diversity of their gut microbiota. Wei et al. [74] reported that higher α-diversity is one of the indicators of improved growth performance in pigs. Our results showed that dietary supplementation with 2% FCHs significantly increased colonic microbiota richness and Shannon and ACE indices. Bacteroides in the intestinal tract can use polysaccharides, which can enhance the host’s nutrient absorption capacity and help the formation of intestinal mucosa [75]. Although previous studies have shown that obesity is associated with a high ratio of Firmicutes to Bacteroidetes [76], the interaction between host nutrient absorption and gut microbial composition is extremely complex and not directly related to the ratio of Firmicutes to Bacteroidetes [77]. Rajput et al. [78] found that the IgA level was negatively correlated with the ratio of Firmicutes to Bacteroidetes. The IgA level was negatively correlated with the abundance of Firmicutes and positively correlated with the abundances of Bacteroidetes and Desulfobacterota, which indicates that a higher Firmicutes/Bacteroidetes ratio may be associated with poor mucosal humoral immunity. Otherwise, the decrease in the Firmicutes/Bacteroidetes ratio in the colonic contents increased the serum LZM level in growing pigs [79]. Additionally, Bacteroidetes were more abundant in the intestines of pigs with higher feed efficiency [80,81]. Our results showed that 2% FCH supplementation significantly increased the relative abundance of Bacteroidetes and decreased that of Firmicutes, which might be correlated to the significantly increased levels of serum IgA and LZM.

*Prevotella* is one of the most important genera in the gut of growing pigs [82] and is positively associated with the host’s high cellulose intake, polysaccharide metabolism [83], feed efficiency [84], and body weight gain [85]. *Phascolarctobacterium* is a gram negative anaerobic bacterium that degrades dietary fiber [86,87]. Its decreased abundance is usually associated with inflammatory diseases, and its increased abundance is beneficial in reducing the occurrence of colitis [88]. *Quinella* is a common propionate-producing bacterium in the gut that plays a role in host nutrient uptake and utilization [89]. *Sphaerochaeta* is an intestinal short-chain fatty acid (SCFA)-producing bacterium that is highly enriched in genes involved in fermentation and carbohydrate metabolism [90]. It produces a variety of carbohydrate enzymes (for example, β-xylosidases of the GH family process carbohydrates), rendering it an important contribution to lipid regulation in the body [91]. *Anaerovibrio*, *Fibrobacter*, and *Treponema* process indigestible polysaccharides into SCFAs [92,93,94,95]. *Oscillospira* participates in glucose synthesis and lipid metabolism in the host’s body and its abundance is positively correlated with propionate production. It is a common core bacterium related to body health in animal intestinal microbiota [96,97]. *Parabacteroides* is a gram negative bacterium that plays an important role in anti-inflammatory and lipid metabolism in the host [98]. Pedersen et al. [99] found a higher relative abundance of *Prevotella* and *Bacteroides* in the gut of pigs with higher body weights. Our results showed that supplementation with 2% FCHs significantly increased the relative abundance of the cellulose-degrading and SCFA-producing bacteria *Parabacteroides* and *Treponema* in the colon microbiota of growing pigs. The correlation pattern between these bacteria and the gut-content metabolites was almost opposite to the pattern between the gut-content metabolites and multiple opportunistic pathogens, such as *Escherichia_Shigella* [100], *Streptococcus* [101,102], *Terrisporobacter* [103], and *Clostridium sensu stricto1* [104]. Dietary supplementation with 2% FCHs significantly reduced the relative abundance of these opportunistic pathogens.

Inflammation caused by *Clostridium sensu stricto* 1 reduces the SCFA content in the gut [105]. The *Eubacterium ruminantium* group comprises potentially pro-inflammatory gut microbiota [106]. *Terrisporobacter* produces trimethylamine-N-oxide, which induces intestinal oxidative stress and inflammation in the host [107] and is highly associated with sepsis [108]. Our results indicated that adding 2% FCHs to the diet significantly reduced the relative abundance of these bacteria, indicating that FCHs inhibited the colonization of pro-inflammatory bacteria in the gut and may help the body resist inflammation.

SDA is a class of polyunsaturated fatty acids that are intermediates in the endogenous conversion of α-linolenic acid to eicosapentaenoic acid [109]. SDA has many beneficial effects, such as regulating lipid metabolism [110] and demonstrating anti-inflammatory [111] and antitumoral effects [112]. Wu et al. [113] reported that the intake of foods rich in SDA could reduce the production of PGE2, which showed a significant pro-inflammatory effect. Propionic acid, an SCFA, is derived from the intestinal anaerobic fermentation of dietary fiber and complex polysaccharides and has beneficial properties in different disease conditions, such as intestinal development and cardiometabolic diseases [114]. SCFAs are common fiber degradation products that can be used as signal metabolites and energy substances for microorganisms to help the host stabilize metabolism and intestinal health [115]. In experiments in apolipoprotein E-deficient (ApoE^−/−^) mice with hypercholesterolemia and atherosclerosis susceptibility induced with a high-fat diet, Haghikia et al. [116] identified a novel regulatory pathway in which propionate acts as a prebiotic to selectively regulate the immune system in the gut, resulting in a reduction in the aortic atherosclerotic lesion area and the control of intestinal cholesterol homeostasis. In this study, supplementation with 2% FCHs significantly increased the colonic contents of SDA and propionic acid in growing pigs.

## 5. Conclusions

The addition of 2% or 3% FCHs significantly improved the growth performance, serum antioxidant, and immunity of growing pigs. In the colon, the dietary supplementation of 2% FCHs significantly increased the relative abundance of cellulose fermentation and SCFA-producing bacteria, decreased the relative abundance of opportunistic pathogens, and significantly increased the contents of SDA and propionic acid. These results indicated that FCHs were a good feed additive for growing pigs, and the recommended addition ratio was 3%. Further study is needed to investigate the effects of higher levels of FCHs on the growth and immunity of growing pigs.

## Figures and Tables

**Figure 1 animals-13-03867-f001:**
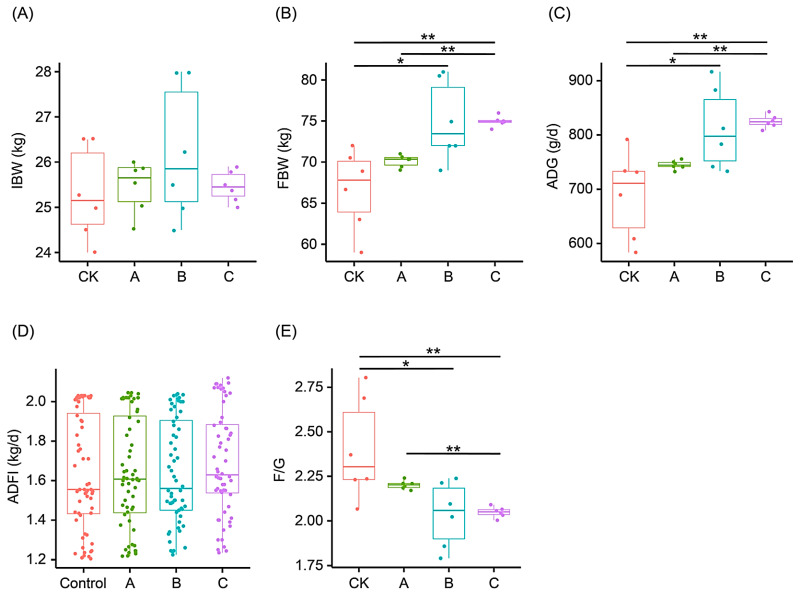
Effects of fermented Chinese herbs (FCHs) on the growth performances of growing pigs. (**A**) Initial body weight (IBW); (**B**) final body weight (FBW); (**C**) average daily gain (ADG); (**D**) average daily feed intake (ADFI); (**E**) feed/gain ratio (F/G). Pigs in the control group were fed a basal diet, and the diets of groups A, B, and C were a basal diet with 1%, 2%, and 3% FCHs, respectively. * *p* < 0.05; ** *p* < 0.01.

**Figure 2 animals-13-03867-f002:**
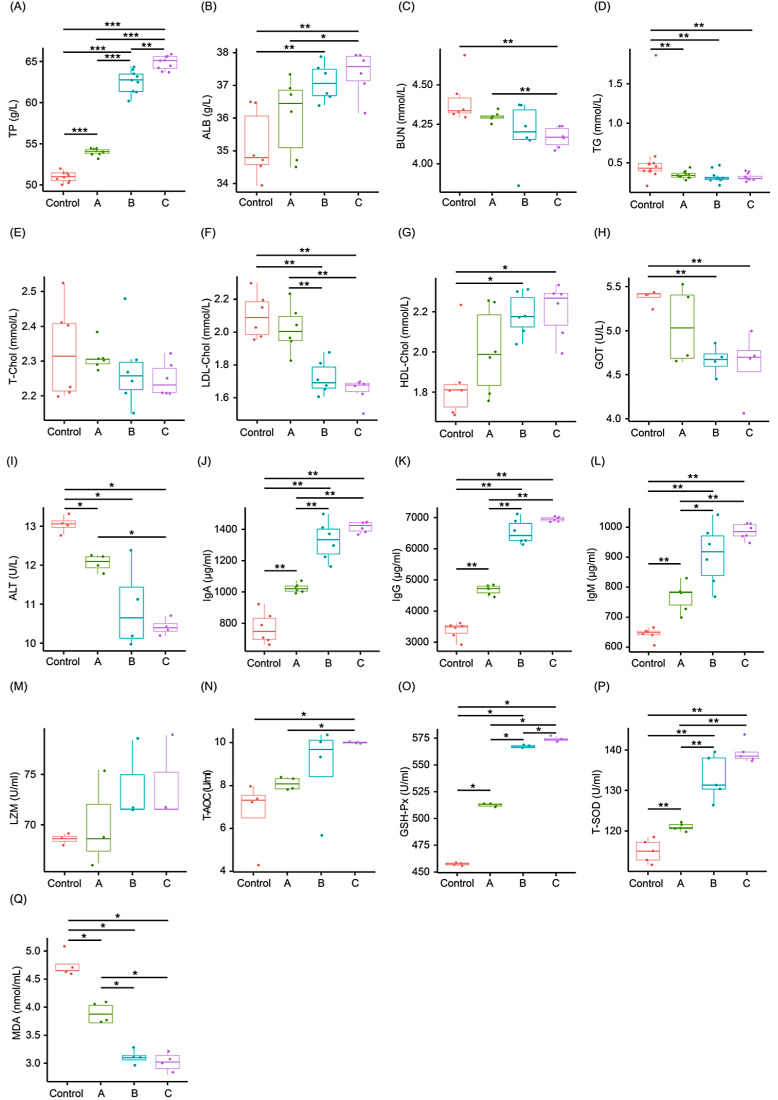
Effects of fermented Chinese herbs (FCHs) on the serum physiological and biochemical indicators of growing pigs. (**A**) Total protein (TP); (**B**) albumin (ALB); (**C**) blood urea nitrogen (BUN); (**D**) triglyceride (TG); (**E**) total cholesterol (T-Chol); (**F**) glutamate oxaloacetate transaminase (GOT); (**G**) alanine aminotransferase (ALT); (**H**) low-density lipoprotein cholesterol (LDL-Chol); (**I**) high-density lipoprotein cholesterol (HDL-Chol); (**J**) total antioxidant capacity (T-AOC); (**K**) glutathione peroxidase (GSH-Px); (**L**) total superoxide dismutase (T-SOD); (**M**) malondialdehyde (MDA); (**N**) immunoglobulin A (IgA); (**O**) immunoglobulin G (IgG); (**P**) immunoglobulin M (IgM); (**Q**) lysozyme (LZM). Pigs in the control group were fed a basal diet, and the diets in groups A, B, and C were provided with a basal diet with 1%, 2%, and 3% FCHs, respectively. * *p* < 0.05; ** *p* < 0.01; *** *p* < 0.001.

**Figure 3 animals-13-03867-f003:**
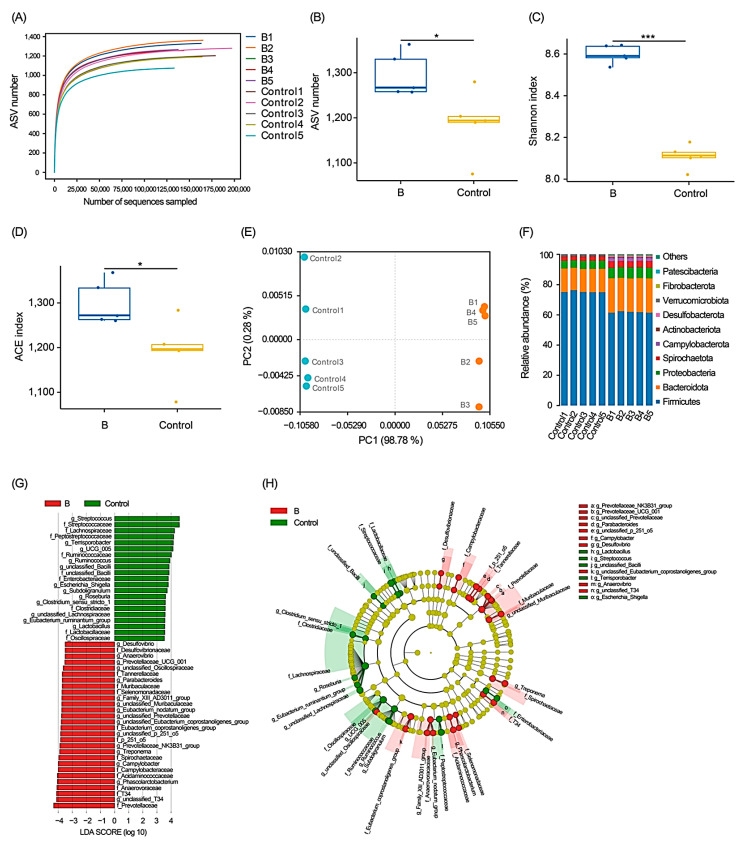
Effect of fermented Chinese herbs (FCHs) on the colonic microbiota of growing pigs. (**A**) Rarefaction curve; (**B**) ASV number; (**C**) Shannon index; (**D**) ACE index; (**E**) principal co-ordinates analysis profile; (**F**) relative abundances of dominant phyla; (**G**) bar chart showing the linear discriminant analysis (LDA) results; (**H**) cladogram plot showing the LDA effect size (LEfSe) results. Taxa with LDA > 3.0 and *p* < 0.05 were considered significant differences; the prefixes “f” and “g” represent the annotated family and genus levels, respectively. Pigs in the control group were fed a basal diet, and the diet of group B was the basal diet with 2% FCH. * *p* < 0.05; *** *p* < 0.001.

**Figure 4 animals-13-03867-f004:**
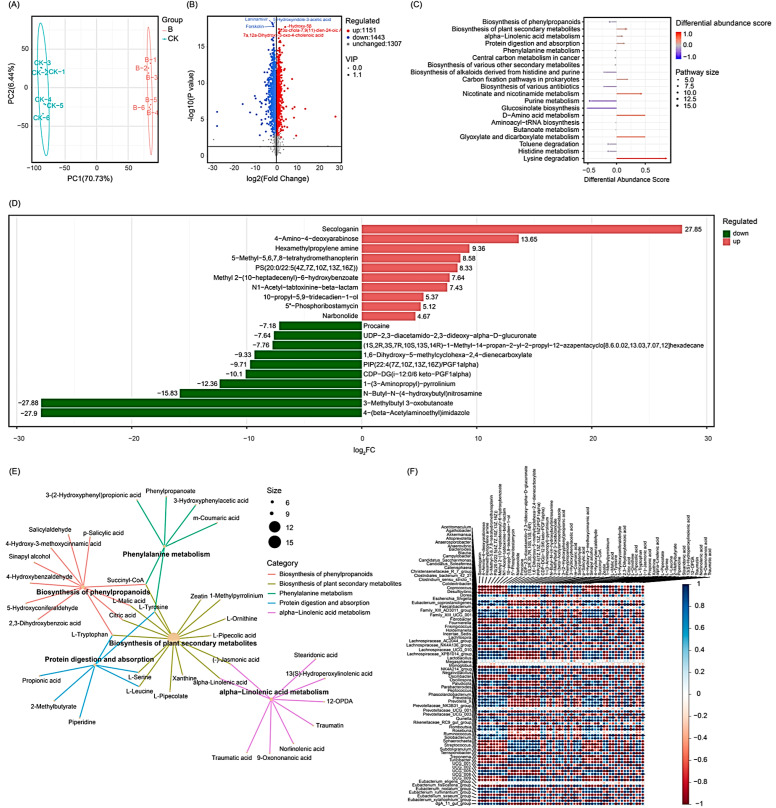
Effect of fermented Chinese herbs on the metabolomics of the colonic contents of growing pigs. (**A**) The principal component analysis profile shows the difference in the metabolomes of the colonic contents between group B and the controls; (**B**) the volcano plot shows the different metabolites; (**C**) the bar chart shows the main differential metabolic pathways; (**D**) the bar chart shows the main differential metabolites; (**E**) the co-occurrence network shows the connections of the main metabolites and metabolic pathways; (**F**) the heatmap profile shows the collection between the main colonic microbes and metabolites. The enlarged heatmap profile is shown in Appendix A. Pigs in the control group were fed a basal diet, and the diet of group B was the basal diet with 2% FCH. * *p* < 0.05; ** *p* < 0.01; *** *p* < 0.001.

**Table 1 animals-13-03867-t001:** Ingredient and nutrient components of the basal diet (air-dry basis) used in the study.

Ingredients	Content (%)	Nutrient Components	Content
Corn	75.45	Metabolizable energy (MJ/kg) ^2^	13.59
Soybean meal	20	Crude protein (%)	15.37
Calcium monophosphate	1.2	Crude fiber (%)	4.15
Shell powder	0.8	ADF (%) ^3^	5.13
Salt	0.35	NDF (%) ^4^	18.64
Choline chloride	0.2	Ash content (%)	4.26
Soybean oil	1.0	Calcium (%)	0.62
Premix ^1^	1.0	Total phosphorus (%)	0.39
Total	100.0		

^1^ The premix provides 110 mg of iron, 5.5 mg of copper, 50 mg of zinc, 0.35 mg of selenium, 10 mg of manganese, 22 mg of pantothenic acid, 0.02 mg of folic acid, 18.5 mg of niacin, 500 IU of vitamin A3, 10 mg of vitamin B1, 20 mg of vitamin B2, 12 mg of vitamin B6, 0.05 mg of vitamin B12, 500 IU of vitamin D, 45 IU of vitamin E, 2.5 mg of vitamin K3, and 0.003 mg of biotin per kilogram of basal diet. ^2^ The metabolizable energy datum was calculated, and the other nutrient components were actual measured values. ^3^ ADF, acid detergent fiber. ^4^ NDF, neutral detergent fiber.

## Data Availability

The raw sequences were deposited in the NCBI Sequence Read Archive database with accession number PRJNA1031988.

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
