# Peer review of "Fermented Chinese Herbs Improve the Growth and Immunity of Growing Pigs through Regulating Colon Microbiota and Metabolites"

_animals, 2023, doi:10.3390/ani13243867_

Round 1

Reviewer 1 Report

Comments and Suggestions for Authors

Author Response

Comments and Suggestions for Authors

Comment

The manuscript titled “Fermented Chinese herbs improve growth and immunity of 2 growing pigs through regulating colon microbiota and metabolites” by Junhao Zhang, Yi Zhou, Chao Hu, Shile Lan provide a valuable in-depth description of the impact that dietary fermented Chinese herb (FCH) on growing pigs performance, biochemical markers, colon microbiota, and metabolome of colon contents. The authors recommend a level of 2 or 3% FCH in pigs’ diets based on the findings they got. However, there are no specifics provided on economic efficiency for a practical standpoint.

Response

Thank you very much for your comment. The purpose of this study was to assess the effects of fermented Chinese herbs on the growth and immunity of growing pigs. As fundamental exploratory research, we purchased relatively few Chinese herbs at higher prices compared to bulk purchases, so we cannot comprehensively evaluate the economic benefits of using the fermented Chinese herbs.

Comments

The authors’ investigation of the effects of FCH on the gut microbiota and gut physiology, including immune function, makes the topic relevant to the goal of the journal’s special issue on ANIMALS.

The references are relevant for the topic.

Response

Thank you for your comment. We fully agree with your comments.

Specific comment

Introduction

Line 57-58: Please use the detailed name before abbreviations or symbols ……

Response

Thank you for your comment. We have added the detailed names according to your comment.

Specific comment

Table 1. Please refer to the premix components, abbreviations and any other explanations bellow the Table. Just the title and number should remain above the Table.

Response

Thank you for your comment. We have moved the premix components, abbreviations and any other explanations bellow the Table according to your comment.

Specific comment

Material and Method

2.4 Determination of serum immune, antioxidant, and biochemical indicators

Describe the method in the order mentioned in the title of the subsection and follow the same order throughout the manuscript.

Response

Thank you for your comment. We have revised our manuscript according to your comment.

Specific comment

2.7 Data analysis: I would suggest using SEM instead of ±SD

Response

Thank you for your comment. We have used the standard error instead of standard deviation according to your comment.

Specific comment

Results

I would suggest to follow the same sections as for Material and Method to be easier to follow and understanding.

Response

Thank you for your comment. We have revised our manuscript according to your comment.

Specific comment

Figures: Figures are hard to understand, especially Picture 2. Please enlarge them! Avoid to use ±SD into the Figures. Re-arrange them: eg. T-Chol whitin the same line with LDL-Chol and HDL-Chol (use the same abbreviation for Cholesterol, please!

Response

Thank you for your comment. We have revised the figures according to your comment.

Specific comment

Line 222. the detailed name before the abbreviation please

Response

Thank you for your comment. We have added the detailed name according to your comment.

Specific comment

I recommend this paper for publication.

Response

Thank you for your recommendation.

Reviewer 2 Report

Comments and Suggestions for Authors

The paper entitled “Fermented Chinese herbs improve growth and immunity of growing pigs through regulating colon microbiota and metabolites” studied the fermented Chinese herbs (FCHs) on the growth and immunity of growing pigs. This study concluded that the FCHs were a good feed additive for growing pigs. This study has certain practical significance.

L38-39, “the recommended addition ratio was 2% – 3%”, this conclusion lacks sufficient data support as the most ratio was 3% in this paper. According to the serum physiological and biochemical indicators of Figure 2, the immune indexes values of group C were higher than group B.

L245, Figure 3G-3H The figures were obscure.

L305, Figure 4F The figure was very obscure.

L339-L342, Description was difficult to understand, is this your conclusion? References were not necessary.

L439-L440, No accurate research or data were found in this paper.

L447-L448, This conclusion lacks sufficient data support.

Comments on the Quality of English Language

Minor editing of English language required

Author Response

Comment

The paper entitled “Fermented Chinese herbs improve growth and immunity of growing pigs through regulating colon microbiota and metabolites” studied the fermented Chinese herbs (FCHs) on the growth and immunity of growing pigs. This study concluded that the FCHs were a good feed additive for growing pigs. This study has certain practical significance.

Response

Thank you very much for reviewing our manuscript and providing very helpful comments.

Comment

L38-39, “the recommended addition ratio was 2% – 3%”, this conclusion lacks sufficient data support as the most ratio was 3% in this paper. According to the serum physiological and biochemical indicators of Figure 2, the immune indexes values of group C were higher than group B.

Response

Thank you for your comment. We have deleted the 2%. Although we did not analyze the effects of higher levels of FCHs on the growth and immunity of growing pigs, our results indicated that addition of 2% or 3% FCHs significantly improved the growth performance and serum antioxidant and immune in growing pigs. Therefore, we believe that the conclusion is still valid. However, further study is need to invest the effect of higher levels of FCHs on the growth and immunity of growing pigs.

Comment

L245, Figure 3G-3H The figures were obscure.

Response

Thank you for your comment. We have enlarged the figure.

Comment

L305, Figure 4F The figure was very obscure.

Response

Thank you for your comment. The enlarged heatmap profile was shown in the Figure S3.

Comment

L339-L342, Description was difficult to understand, is this your conclusion? References were not necessary.

Response

It’s our conclusion. We have revised the sentence.

Comment

L439-L440, No accurate research or data were found in this paper.

Response

Thank you for your comment. We have deleted the sentence according to your comment.

Comment

L447-L448, This conclusion lacks sufficient data support.

Response

Thank you for your comment. Although we did not analyze the effects of higher levels of FCHs on the growth and immunity of growing pigs, our results indicated that addition of 2% or 3% FCHs significantly improved the growth performance and serum antioxidant and immune in growing pigs. Therefore, we believe that the conclusion is still valid. However, further study is need to invest the effect of higher levels of FCHs on the growth and immunity of growing pigs.

Comments on the Quality of English Language

Minor editing of English language required

Response

We have checked our manuscript and revised language errors according to your comment.

Reviewer 3 Report

Comments and Suggestions for Authors

The article is well written and has scientific merit.

However, some questions need to be answered, such as:

 - There is no justification for the fermentation of Chinese herbs in the introduction. Why ferment? What are the expected benefits? What is expected to happen to the secondary metabolism of each plant?

 -In the methodology, how long was the fermented product dried?

 -There was some analysis of the secondary metabolite profile before and after fermentation and drying.

 -The inclusion rate of each plant for fermentation was also unclear.

Author Response

Comment

The article is well written and has scientific merit.

However, some questions need to be answered, such as:

Response

Thank you very much for reviewing our manuscript and providing very helpful comments. We have revised our manuscript according to your comments.

Comment

 - There is no justification for the fermentation of Chinese herbs in the introduction. Why ferment? What are the expected benefits? What is expected to happen to the secondary metabolism of each plant?

Response

Chinese herbs have been used to cure diseases for thousands of years. However, the bioactive ingredients of Chinese herbs are complex, and some Chinese herb natural products cannot be directly absorbed by human and animals. Moreover, the contents of most bioactive ingredients in Chinese herbs are low, and some natural products are toxic to humans and animals. Fermentation is a Chinese herb processing technique based on the action of microorganisms under the proper temperature, humidity, and moisture conditions that enhances their original properties and/or produces new effects, expanding their scope of application to meet clinical demands. Furthermore, Fermentation of Chinese herbs could enhance Chinese herb bioactivities and decrease the potential toxicities. We have added the description in the Introduction section of our revised manuscript.

Comment

 -In the methodology, how long was the fermented product dried?

Response

The fermented product was dried approximately 4 hours at 60 °C to 10% moisture. We have added the dried time in our revised manuscript.

Comment

 -There was some analysis of the secondary metabolite profile before and after fermentation and drying.

Response

Thank you for your comment. We regret that we did not analyze the secondary metabolites before and after fermentation and drying.

Comment

 -The inclusion rate of each plant for fermentation was also unclear.

Response

Thank you for your comment. As we description in the Materials and Methods, mix dried hawthorn (Crataegus pinnatifida Bunge), dried tangerine peel, mother-wort herb (Leonurus japonicus Houtt.), dried Magnolia officinalis bark, Radix Astragali, Radix Cynanchum otophyllum Schneid., folium artemisiae argyi, licorice (Glycyrrhiza uralensis Fisch.) root, and dandelion (Taraxacum mongolicum Hand.-Mazz.) in a ratio of 3 : 4 : 1 : 1 : 2 : 2 : 4 : 1 : 1 was crushed and passed through a 60 mesh sieve, adjusted the water content to 40%, and inoculated Bacillus subtilis (viable count ≥ 2.5×109 CFU/g), then placed in a fermentation bag, fermented for 5 days in an incubator with 37 °C, and finally dried at 60 °C.